# Accuracy of hepatitis B disease surveillance, Gannan prefecture, Gansu province, China; 2017

**Pinggui Wang**[1], **Lijie Zhang**[2]*, **Jian He**[1¤]*, **Guomin Zhang**[3], **Yvan Ma**[4], **Weimin Lv**[4], **Xiaoshu Zhang**[1], **Jin An**[1]

**1** Department of Expanded Programme on Immunization, Gansu Center for Disease Control and Prevention, Lanzhou, China, **2** Department of Education and Training, Chinese Center for Disease Control and Prevention, Beijing, China, **3** National Immunization Program, Chinese Center for Disease Control and Prevention, Beijing, China, **4** Department of Expanded Programme on Immunization, Gannan Center for Disease Control and Prevention, Hezuo, Gannan, China

¤ Current address: Gansu Provincial Red Cross Blood Center, Lanzhou, China
* zhanglj@chinacdc.cn (LZ); 1014770674@qq.com (JH)

**Data Availability Statement:** All relevant data are within the paper and its Supporting Information files.

**Funding:** This project was funded by Chinese Field Epidemiology Training Program of Chinese Center

## Abstract

Hepatitis B is a major global public health threat. According to China's National Notifiable Disease Reporting System (NNDRS), Gannan Tibetan Autonomous Prefecture (Gannan) had the highest incidence of hepatitis B in Gansu Province during 2004 to 2016. We evaluated NNDRS hepatitis B case reports from Gannan to determine accuracy of diagnosis and to understand factors associated with inaccuracy. We reviewed medical records with hepatitis B diagnosis hospitalized in seven county hospitals in Gannan between January 1, 2016 and July 31, 2017. Using national "Classification and Diagnostic Procedures for Hepatitis B," we independently reclassified the diagnoses. We determined the positive predictive value (PPV) of reported hepatitis B cases. We investigate clinicians' understanding of the diagnostic and reporting criteria for hepatitis B by questionnaire. We reviewed and re-categorized 400 inpatients reported. Sixteen cases had been reported as acute hepatitis B, but on re-categorization, none were acute hepatitis B cases. PPVs for chronic hepatitis B and unclassified hepatitis B cases were 66% and 15% respectively; 327 (82%) of the reported hepatitis B cases were inaccurately classified; 261 were carriers, 59 were reported previously, and 7 did not have hepatitis B. The actual incidence of hepatitis B in Gannan in 2016 was estimated to be 19/100,000, significantly below the reported incidence of 106/100,000. Among reported cases, 81% had been tested for Alanine aminotransferase, 52% for hepatitis C antibody, 80% with liver ultrasound, 32% for hepatitis A antibody, and 7% for hepatitis B virus (HBV) DNA. Not all cases were tested for anti-HBc IgM or hepatitis E antibody or had a liver biopsy. In the knowledge test, 56% of clinicians accurately diagnosed three simulated cases of acute hepatitis B, and 17% correctly diagnosed two simulated cases chronic hepatitis B; 22% knew that "a client with only HBsAg positivity need not be reported." The falsely high incidence in Gannan was due to diagnostic and reporting inaccuracies. We recommend that clinicians and laboratorians receive additional training in hepatitis B diagnostic criteria and reporting standards,

for Disease Control and Prevention. Chinese field epidemiology training program reviewed this study design and provided the fund suppprting for this study. The project number is 131031001000190032-719025. Teachers of the training program acted as co-authors had role in study design, data analysis, and preparation of the manuscript.

**Competing interests:** he authors have declared that no competing interests exist.

including appropriate use of IgM anti-HBc tests. Hepatitis B surveillance data should be periodically reviewed and evaluated for accuracy.

## Introduction

Hepatitis B virus (HBV) is a major global health threat due to its ease of vertical and horizontal transmission and widespread prevalence. According to the World Health Organization (WHO), In 2019, hepatitis B resulted in an estimated 820 000 deaths, mostly from cirrhosis and hepatocellular carcinoma (primary liver cancer). 296 million people around the world are living with HBV (are hepatitis B surface antigen [HBsAg] positive). The prevalence of HBV infection is the highest in the Western Pacific Region [1] - 116 million people in the Western Pacific Region and 90 million people in China are living with HBV.

The government of China adopted increasingly comprehensive strategies to prevent HBV transmission, including immunization, promotion of safe injection practices, blood donation screening, and surveillance [2]. Recombinant hepatitis B vaccine (HepB) received market authorization in 1992 and was placed in the management of the National Immunization Program and recommended for newborns and infants. In 2002, China integrated HepB into its Expanded Program on Immunization (EPI), making the vaccine available at no cost to children through 14 years of age [3, 4]. Compared with a 1992 serological survey, overall HBV surface antigen prevalence of persons 1–29 years of age declined 46% by 2006 and 52% by 2014 [5].

China has not established a standalone surveillance system for hepatitis B, relying instead on the decades old National Notifiable Disease Reporting System (NNDRS) for hepatitis B surveillance. In 2004, China developed a disease control network with direct reporting of hepatitis B cases categorized by reporting clinicians as acute or chronic according to *the Diagnostic Criteria of Viral Hepatitis B*. Reports from this network are transferred to NNDRS. Classification should be based on clinical symptoms, signs, infection times, laboratory testing, and auxiliary examination results [6]. Cases that cannot be classified are reported as "unclassified hepatitis B cases" and considered as classification failures of the surveillance system. CHB cases should be reported to the surveillance system only one time to avoid duplicate counting [7].

There are concerns about the accuracy of hepatitis B cases reported to NNDRS. For example, an eight-province study in China by Wang and colleagues found that only 35% of cases reported as acute hepatitis B (AHB) in 2007 were able to be verified by chart review as actual AHB cases [8]. The investigators also found that the proportion of unclassified hepatitis B cases was large, demonstrating the need for more discriminant diagnostic criteria. In 2010, 36% of hepatitis B cases reported nationwide were unclassified [9]. Additional concerns have been that hepatitis B carriers identified during routine checkups or by preoperative testing are often reported as hepatitis B cases, and CHB cases have been repeatedly reported as a primary cases [10]. Taken together, these issues result in reported incidences of new hepatitis B infections being higher than the true incidences.

Although hepatitis B vaccination of newborns, infants, and children has dramatically decreased perinatal transmission and led to a 97% reduction in HBsAg positivity among children under 5 years of age [5], overall, all-age rates of hepatitis B have increased steadily in China between 1990 and 2008. Experts have postulated that part of this paradox may be due to failure to distinguish acute from chronic HBV infections [11]. To address this question and provide scientific evidence supporting effective hepatitis B preventive and control measures, we conducted an evaluation of the accuracy of NNDRS hepatitis B case reports from an area

with a high incidence of hepatitis B. We report results of our evaluation and make recommendations for improving surveillance accuracy.

## Methods

### Setting

The study setting is Gannan Tibetan Autonomous Prefecture (Gannan) in Gansu Province. Gannan is located on a plateau with an area of 45,000 square kilometers and is one of ten Tibetan autonomous states in China. Gannan has a population of 730,000 people, 54% of which are Tibetan. According to 2016 NNDRS data, the incidence of hepatitis B was 69/100,000 in China, 38/100,000 in Gansu Province, and 106/100,000 in Gannan. From 2004 through 2015, Gannan's hepatitis B incidence was significantly higher than provincial and national average levels.

In 2016, the 8 counties in Gannan prefecture reported a total of 745 hepatitis B cases. Counties with the most reports were Luqu County, Xiahe County, Zhouqu County, and Zhuoni County—586 cases were reported from these counties, accounting for nearly 80% of the all reports from Gannan. Nine county-level hospitals in these four counties reported 95% of cases; we included seven of these hospitals in our study.

### Case classify and diagnose

According to the National diagnosis and reporting guideline of hepatitis B, Cases with hepatitis B symptoms and/or ALT abnormalities have HBsAg positive, if the HBsAg positive result was tested within 6 months, the case should be diagnosed as acute hepatitis B. If the HBsAg positive result was tested 6 months ago, the case should be diagnosed as chronic hepatitis B. When the tested time of the HBsAg positive result was unknown, Acute hepatitis B can be diagnosed if the anti-HBc IgM was positive. If a case with HBsAg positive test results had no hepatitis B symptoms and no ALT abnormalities, the case should be diagnosed as hepatitis B carrier.

### Case reviews

We obtained medical records of hepatitis B patients hospitalized between January 1, 2016 and July 31, 2017, including admission examination results, past medical histories, family histories, laboratory examinations, and B-ultrasound examinations. We obtained case report data from NNDRS, including demographic data and onset, diagnosis, and reporting dates. Using the "Classification Diagnostic Process of Hepatitis B cases" issued by China CDC in 2012, we reevaluated each case for accuracy of diagnosis. Using our case evaluations as the gold standard, we determined the positive predictive value (PPV) of hepatitis B reports to NNDRS. We considered PPV to be the proportion of the true hepatitis B cases among reported cases.

### Clinician interviews

We evaluated clinicians' diagnosis capacity and accuracy of reporting for hepatitis B by interviewing using a questionnaire. We interviewed all the clinicians who were working in the 7 hospitals where cases were reported when the survey was conducted. We set 6 simulated case scenarios to identify the accuracy of diagnosis of hepatitis B for clinicians. Scenario 1: a patient has been confirmed HBsAg negative in the previous six months, but became HBsAg positive with associated signs and symptoms of liver disease or with an abnormal alanine aminotransferase (ALT) test. Scenario 2: a patient whose previous HBsAg test results are not available now has a positive HBsAg test result, with associated signs and symptoms. In the recovery period, serum HBsAg tested negative and anti-HBs was positive. Scenario 3: a patient had no

previous HBsAg tests, but now is positive for HBsAg and has an abnormal ALT; liver biopsy shows changes consistent with acute hepatitis. Scenario 4: a patient whose is known to be HBsAg positive for more than 6 months and has signs and symptoms associated with CHB and has ALT abnormalities for the first time. Scenario 5: a patient has not been tested for HBsAg before and now tests positive for HBsAg and has associated symptoms, an abnormal ALT, and a negative anti-HBc IgM test. Scenario 6: a patient has not been tested for HBsAg before and now has a positive HBsAg test with no related signs or symptoms and a normal ALT test. The six simulated case scenarios included three AHB case scenarios, two CHB case scenarios, and one hepatitis B virus carrier scenario. Scenario 1 to 3 were for CHB—hepatitis B surface antigen detection is positive, the course of disease has lasted more than half a year or the date of onset is not clear and a patient has clinical manifestations of chronic hepatitis; Scenario 4 and 5 were for AHB—a newly developed inflammation of the liver caused by hepatitis B virus infection; and scenario 6 was for hepatitis B virus carriers—hepatitis B virus (HBV) infections with HBsAg positivity for more than 6 months, with few symptoms or signs related to liver disease, and with normal liver function. We set 1 point for each scenario if diagnosis correctly. We calculated the mean scores among 41 clinicians for the 6 scenarios.

There were 5 questions in the questionnaire to assess weather the clinicians understood the reporting criteria correctly for acute and chronic hepatitis B in the NNDRS.

## Ethics statement

Ethics approval was obtained for this study by Ethical Review Committee in Gansu Center for Disease Control and Prevention. Consent for participate was obtained from each participant.

## Results

### Diagnostic reassessment

A total of 799 cases of hepatitis B was reported from the seven hospitals; 427 (53%) were inpatients and 372 (47%) were outpatients. Among the inpatients, 400 (94%) were reassessed for accuracy of diagnosis; 16 (4%) had been reported as AHB; 199 (50%) had been reported as CHB; 185 (46%) were reported unclassified.

Upon re-evaluation of the 16 cases reported as acute hepatitis B, there were two previously unreported CHB cases and no true AHB cases. The 14 cases that should not have been reported included eight repeated reports of CHB and six HBV carriers (Table 1); 88% (14/16) of reported AHB cases were inaccurately classified and reported. The positive predictive value of reported AHB was 0% (0 /16).

Among 199 cases reported as CHB, eleven were reassessed to be true CHB cases, and 28 were reassessed as unclassifiable hepatitis B cases. The remaining 160 cases should not have been reported, and among these were 119 hepatitis B carriers, 37 duplicate reports of CHB cases, and four individuals that did not have hepatitis B (one person with chronic

**Table 1. Evaluation of 400 hepatitis B cases using standardized China CDC diagnostic criteria in Gannan from 2016 to 2017.**

| Reported diagnosis | Cases reported to NNDRS | Re-classification by CCDC guidelines for Diagnostic classification of Hepatitis B cases | | |
|---|---|---|---|---|
| | | Acute | Chronic | Unclassified |
| Acute | 16 | 0 | 2 | 0 |
| Chronic | 199 | 0 | 11 | 28 |
| Unclassified | 185 | 0 | 5 | 27 |
| Total | 400 | 0 | 18 | 55 |

gastroenteritis, one with dermatitis, one with head trauma, and one with costal neuritis). In all, 80% (160/199) of reported CHB cases were inaccurately classified. The positive predictive value of reported CHB was 6% (11/199).

Of 185 cases reported as unclassified hepatitis B, 27 were true unclassified hepatitis B cases and 5 were CHB cases. The remaining 153 were inaccurately classified, including 136 hepatitis B carriers, 14 duplicate case reports of CHB, and individuals that did not have hepatitis B (one pregnant woman who was HBsAg negative, one person with hepatitis A, and one person with hepatitis C). In all, 83% (153/185) of reported unclassified cases should not have been reported. The positive predictive value of reported unclassified hepatitis B cases was 15% (27/185).

Among the 400 reported hepatitis B cases in this survey, after reevaluation and reclassifying, only 73 (19%) should have been reported to NNDRS—18 chronic hepatitis B cases and 55 unclassified hepatitis B cases. The 55 cases of unclassified hepatitis B had no further testing while in the hospital and therefore could not be classified as AHB or CHB. Among the remaining 327 (82%) cases of hepatitis B, 261 were hepatitis B carriers, 59 were duplicate reports of CHB, and seven reported individuals did not have hepatitis B. Using the reclassified cases instead of the reported cases in Gannan, the true incidence of hepatitis B was 19/100, 000; 82% of reported cases were inaccurately reported. The positive predictive values among reported acute hepatitis B cases, reported chronic hepatitis B cases and reported unclassified hepatitis B cases were 0%(0/16), 5.5%(11/199) and 14.6%(27/185) (Table 1).

We calculated the inaccurately reporting rates in different departments. The highest inaccurately report rate were 100% in pediatrics, followed by obstetrics and gynecology (91%), traditional Chinese medicine (81%), surgery(74%) and internal medicine(74%). Among 59 duplicated reports, 93%(55/59) were reported from internal medicine department. Among 261 hepatitis B carriers, 47%(146/261) were reported by gynaecology and obstetrics department.

## Auxiliary hepatitis testing

Among the 400 reported cases of hepatitis B, 81% had ALT testing performed; 80% were tested for hepatitis C antibody; 52% had liver ultrasound; 32% were tested for hepatitis A antibody; and 7% were tested for quantitative HBV DNA. None was tested for anti-HBc IgM, hepatitis E antibodies and liver biopsies. We interviewed 41 clinicians why few IgM tests were performed, all answered this question. 78% (32/41) of clinicians said that this test was not helpful for clinical treatment, and laboratory staff said that they had the ability to test for IgM but lacked the reagents (Table 2).

## Clinician knowledge

We interviewed 41 clinicians who worked at the 7 hospitals where the diagnosis of hepatitis B was surveyed. There were 20 males and 21 females. 36 of them were college and bachelor

**Table 2. Laboratory diagnostic tests in 400 cases of hepatitis B reported in Gannan.**

| Lab exam | Cases number (%) |
|---|---|
| ALT | 325 (81) |
| Anti-HCV | 318 (80) |
| Liver B ultrasound | 206 (52) |
| Anti-HAV | 128 (32) |
| HBV DNA | 28 (7) |
| Anti-HBc IgM($>$1:1000 is positive) | 0 (0) |
| liver needle biopsy | 0 (0) |
| Anti-HEV | 0 (0) |

**Table 3. Knowledge of the diagnostic criteria of hepatitis B among 41 clinicians in Gannan prefecture of Gansu province in 2017.**

| Type of diagnosis | Simulation case | Number of doctors diagnosed correctly | Diagnostic accuracy rate(%) |
|---|---|---|---|
| Acute hepatitis B | Scenorio 1 | 23 | 56 |
| | Scenorio 2 | | |
| | Scenorio 3 | | |
| Chronic hepatitis B | Scenorio 4 | 7 | 17 |
| | Scenorio 5 | | |
| Hepatitis B carriers | Scenorio 6 | 31 | 76 |

degree. 5 of them were high school education level. The professional title of 29 of them were residents, 8 attending physicians, and 4 associate chief physicians. 20 clinicians had less than 10 years of service, 6 had 10–19 years of service, 15 clinicians had more than 20 years of service. 20 of them came from internal department, 15 from obstetrics and gynecology department, the others came from clinical laboratory, department of pediatrics.

Forty-one clinicians participated in the simulated hepatitis B patient questionnaire. We found that 56% (23) of participating clinicians accurately diagnosed the three simulated cases AHB, 17% (7) correctly diagnosed the two simulated cases of CHB, and 76% (31) accurately diagnosed the simulated HBV carriers (Table 3). Among the 41 clinicians, only 3 of them accurately diagnosed all the 6 simulated cases. The mean score was 3.83±2.16 for the 6 scenarios (1 point for each scenario). The mean score was highest in clinicians from department of pediatrics(4.33±2.44), followed by internal medicine department (4.00±1.39), obstetrics and gynecology department (3.73±2.63), the lowest mean score was 2.66±1.85 for clinicians from clinical laboratory department. The mean score of residents, attending physicians, and associate chief physicians, were 3.79 ± 2.08,3.63 ± 2.58, and 4.5 ± 0.98, respectively. The mean score in clinicians with high school degree were 3.40 ± 2.35, college and bachelor were 3.89 ± 2.11. From the length of service of surveyed clinicians, Clinicians with 20 years had the highest score (4.13 ± 1.73), followed by clinicians with less than 10 years (3.85 ± 1.78) and 10–19 years (3.00 ± 3.20).

The investigation results revealed that 22% (9/41) clinicians knew "only HBsAg positive client do not need to be reported", 46%(19/41) clinicians knew "Cases diagnosed as acute hepatitis B need to be reported", 46%(19/41) clinicians knew "The first visit of chronic hepatitis B cases need to report on", 46%(19/41) clinicians knew "Cases previously reported in other hospitals do not need to be reported for the first time in our hospital", 83%(34/41) clinicians knew "Return cases for the first visit of this year also do not need to be reported".

## Discussion

Our study showed that the actual incidence of hepatitis B in Gannan prefecture in 2016 was substantially lower than the incidence reported by Gansu province. We made the decision based on the previous disease history, clinical symptoms, imaging examination, laboratory serological test results recorded in medical records. All these data were accurate and true. So our findings could show the real status of clinicians' diagnosis and reporting of hepatitis B.

There were some reasons for such a high rate of misdiagnosis in the endemic HBV region. First, some clinicians did not understand the diagnosis criteria correctly. The hospital does not organize the training on hepatitis B diagnosis; Second, some testing items conducive to the classified diagnosis of hepatitis B are not carried out in local hospitals, such as Anti-HBc IgM, HBV DNA and liver needle biopsy, perhaps because the classified diagnosis has little significance to the treatment of patients.

A key indicator for distinguishing acute and chronic hepatitis B is the duration of HBsAg positivity [6]. If HBsAg testing is not available, it is recommended to either do an anti-HBc IgM test (>1:1000 is considered positive), perform a liver biopsy, or follow up after 6 months to conduct a proper classification of hepatitis B [12]. Performing a liver biopsy and following up after 6 months may be difficult to do in rural areas.

Anti-HBc IgM testing is also an important method to distinguish acute from chronic cases [11]. Our investigation showed that even though all seven county hospitals in Gannan had the ability to test for anti-HBc IgM, clinicians believed that the test was not helpful for clinical treatment and therefore did not routinely order it. Of the 73 cases of hepatitis B that should have been reported, 55 could not be classified as acute or chronic due to the absence of anti-HBc IgM testing. Although studies have reported that 23.1% of acute CHB flare-ups are IgM positive [13], according to the current diagnostic criteria of hepatitis B in China, clinicians can classify cases according to the anti-HBc IgM test results. If cases can be tested with validated IgM anti-HBc, the diagnostic accuracy of classification of hepatitis B cases could increase by 74%. We recommend training clinicians in diagnostic testing and promoting increased use of anti-HBc IgM testing.

## Study limitations

Our study has limitations to consider when interpreting or generalizing the findings. The study was conducted only among hospitalized cases, precluding generalizing to outpatients, and our clinician sample size was small. However, we believe that our study illustrates problems in the diagnosis and reporting of hepatitis B cases and may help understand reasons for the falsely high incidence of hepatitis B in Gannan.

## Conclusions

The falsely high incidence of hepatitis B in Gannan was due to diagnostic and reporting inaccuracies. The survey concluded that many clinicians in Gannan do not have a clear understanding of the diagnostic and reporting criteria of hepatitis B. Training for medical personnel on the diagnostic criteria and reporting standards of hepatitis B is needed to improve the ability to accurately diagnose and report hepatitis B. Surveillance reports should be routinely reviewed by public health staff.

## Acknowledgments

We are grateful to Dr. Linda Quick in US CDC, Dr. Lance Rodewald, Dr. Fuzhen Wang, Dr. Hui Zheng, Ning Miao, Xiaojin Sun, and Qianli Yuan in China CDC for their critical comments on the manuscript. This work was supported and cooperated by the Gannan prefecture CDC.

## Author Contributions

**Conceptualization:** Jian He.

**Data curation:** Pinggui Wang, Weimin Lv.

**Formal analysis:** Pinggui Wang, Lijie Zhang, Guomin Zhang.

**Funding acquisition:** Lijie Zhang.

**Investigation:** Pinggui Wang, Yvan Ma.

**Methodology:** Pinggui Wang, Jian He, Xiaoshu Zhang, Jin An.

**Project administration:** Pinggui Wang, Yvan Ma, Weimin Lv, Xiaoshu Zhang, Jin An.

**Supervision:** Jian He, Guomin Zhang, Weimin Lv, Xiaoshu Zhang, Jin An.

**Writing – original draft:** Pinggui Wang.

**Writing – review & editing:** Pinggui Wang, Lijie Zhang, Jian He, Guomin Zhang.

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
