## [Decision Letter · Decision Letter 0]

21 Apr 2022

PONE-D-22-06387Accuracy of hepatitis B disease surveillance, Gannan Prefecture, Gansu Province, China; 2017PLOS ONE

Dear Dr. Zhang,

Thank you for submitting your manuscript to PLOS ONE. After careful consideration, we feel that it has merit but does not fully meet PLOS ONE’s publication criteria as it currently stands. Therefore, we invite you to submit a revised version of the manuscript that addresses the points raised during the review process.

We look forward to receiving your revised manuscript.

Kind regards,

Antonio De Vincentis

Academic Editor

PLOS ONE

Journal Requirements:

"This project was funded by Chinese Field Epidemiology Training Program, Chinese Center for Disease Control and Prevention. The project number is 131031001000190032-719025."

We note that you have provided funding information that is not currently declared in your Funding Statement. However, funding information should not appear in the Funding section or other areas of your manuscript. We will only publish funding information present in the Funding Statement section of the online submission form. 

"This study was supported by Chinese Field Epidemiologh Training Program. Project number is 131031001000160016."

Reviewers' comments:

Reviewer's Responses to Questions

**Comments to the Author**

1. Is the manuscript technically sound, and do the data support the conclusions?

Reviewer #1: Partly

Reviewer #2: Yes

2. Has the statistical analysis been performed appropriately and rigorously? 

Reviewer #1: N/A

Reviewer #2: Yes

3. Have the authors made all data underlying the findings in their manuscript fully available?

Reviewer #1: No

Reviewer #2: Yes

4. Is the manuscript presented in an intelligible fashion and written in standard English?

Reviewer #1: Yes

Reviewer #2: Yes

5. Review Comments to the Author

Reviewer #1: In this paper, Wang et al. analyzed the accuracy of diagnosis of hepatitis B in the national diagnosis reporting system in Gannan, China. They concluded that the falsely high incidence in the reporting system was due to diagnostic and reporting inaccuracies. This paper simply shows the poor diagnostic ability of clinicians for hepatitis B in this area. There are some concerns in this paper as below.

1. The reasons why many patients without hepatitis B were overdiagnosed are not fully evaluated. Such data are required to improve the ability of diagnosis.

2. If education for clinicians improves the accuracy of diagnosis of hepatitis B even though partly, it would be important data also for other areas.

3. Page 5, “We interviewed medical staff on factors that affect the quality of hepatitis B detection, diagnosis, and reporting for each of the three categories of HBV infection…” – there is no results of this interview in the result section.

4. Pages 7-8, “When asked by the investigators why few IgM tests were performed…” – please show how many clinicians were asked and how many were answered.

5. Page 8, “Clinician knowledge” – there is little information on the clinicians. Did they work at the 7 hospitals where the diagnosis of hepatitis B was surveyed?

6. Page 8, “Clinician knowledge” – how many clinicians accurately diagnosed all the simulated cases? Additionally, please show mean (or median) score for the 6 scenarios.

Reviewer #2: Zhang and co-authors performed a nice study regarding HBV misdiagnosis. Congratulations for the good idea. The results are really concerning! I have some comments for them:

Can you precisely explain how HBV cases are classified in the NNDRS?

Were the 427 inpatients reported admitted for an HBV related disease? Did outpatients seek medical care for an HBV related disease?

On study limitations you state that “The study was conducted only among hospitalized cases…” but you included 372 outpatients, can you clarify it?

Did you find any similar study in the literature?

I would be nice if you can find any predictor for misclassification such as level of expertise, age, years of practice, subspeciality of clinicians making those reports and for those taking the interviews.

I suggest adding in the discussion section how theses findings may impact in clinical studies performed for data obtained from databases relying only in medical reports not confirmed by serological data.

Why do you think there is such a high rate of misdiagnosis in an endemic HBV region? Please add some comments in the discussion section.

6. PLOS authors have the option to publish the peer review history of their article (what does this mean?). If published, this will include your full peer review and any attached files.

Reviewer #1: No

Reviewer #2: No

---

## [Author Response · Author response to Decision Letter 0]

22 Jun 2022

Response to reviewer #1: 

1. The reasons why many patients without hepatitis B were over diagnosed are not fully evaluated. Such data are required to improve the ability of diagnosis.

In our study, we found that there were 7 inpatients whose hepatitis B surface antigen were negative. By reviewing the medical records of these 7 inpatients, they were diagnosed as chronic gastroenteritis, dermatitis, head trauma, costal neuritis, pregnant woman, hepatitis A and hepatitis C, respectively. They should not be reported to the hepatitis B surveillance system. These 7 inpatients were diagnosed correctly, but some mistakes occurred when the clinician reported to NNDRS. 

2. If education for clinicians improves the accuracy of diagnosis of hepatitis B even though partly, it would be important data also for other areas. 

The survey showed that many clinicians in Gannan didn’t understand the diagnostic and reporting criteria of hepatitis B well which affecting the accuracy of diagnosis of hepatitis B. So providing education for clinicians on the diagnostic and report criteria of hepatitis B is needed to improve the accurate of diagnose and reporting of hepatitis B. We agree that it would be important data also for other areas.

3. Page 5, “We interviewed medical staff on factors that affect the quality of hepatitis B detection, diagnosis, and reporting for each of the three categories of HBV infection…” – there is no results of this interview in the result section.

Sorry for the confusion. We redescribed this sentence to make it more clear. In fact, we evaluated clinicians’ diagnosis capacity and accuracy of reporting for hepatitis B by interviewing using a questionnaire. We interviewed all the clinicians who were working in the 7 hospitals where cases were reported when the survey was conducted. We set 6 simulated case scenarios to identify the accuracy of diagnosis of hepatitis B for clinicians. The six simulated case scenarios included three AHB case scenarios, two CHB case scenarios, and one hepatitis B virus carrier scenario.

We added the results of 41 clinician diagnosis capacity and accuracy of reporting for hepatitis B in the revised manuscript on line 246 in page 10.

4. Pages 7-8, “When asked by the investigators why few IgM tests were performed…” – please show how many clinicians were asked and how many were answered.

We interviewed 41 clinicians and all answered the question. 78% (32/41) of clinicians said that this test was not helpful for clinical treatment, We added this information on line 229 in page 9.

5. Page 8, “Clinician knowledge” – there is little information on the clinicians. Did they work at the 7 hospitals where the diagnosis of hepatitis B was surveyed?

We interviewed 41 clinicians who worked at the 7 hospitals where the diagnosis of hepatitis B was surveyed. There were 20 males and 21 females. 36 of them were college and bachelor degree. 5 of them were high school education level. The professional title of 29 of them were residents, 8 attending physicians, and 4 associate chief physicians. 20 clinicians had less than 10 years of service, 6 had 10-19 years of service, 15 clinicians had more than 20 years of service.20 of them came from internal department, 15 from obstetrics and gynecology department, the others came from clinical laboratory, department of pediatrics. We added this information on line235 in page 9.

6. Page 8, “Clinician knowledge” – how many clinicians accurately diagnosed all the simulated cases? Additionally, please show mean (or median) score for the 6 scenarios.

Among the 41 clinicians, only 3 of them accurately diagnosed all the simulated cases. The mean score was 3.83 ± 2.16 for the 6 scenarios (1 point for each scenario). The mean score was highest in clinicians from department of pediatrics（4.33±2.44）, followed by internal medicine department (4.00±1.39), obstetrics and gynecology department (3.73±2.63), the lowest mean score was 2.66±1.85 for clinicians from clinical laboratory department. The mean score of residents, attending physicians, and associate chief physicians, were 3.79 ± 2.08,3.63 ± 2.58, and 4.5 ± 0.98, respectively. The mean score in clinicians with high school degree were 3.40 ± 2.35, college and bachelor were 3.89 ± 2.11. From the length of service of surveyed clinicians, Clinicians with 20 years had the highest score (4.13 ± 1.73), followed by clinicians with less than 10 years (3.85 ± 1.78) and 10-19 years (3.00 ± 3.20). We added this information on line 246 in page 10.

Response to reviewer #2: 

1.Can you precisely explain how HBV cases are classified in the NNDRS?

According to the National diagnosis and reporting guideline of hepatitis B, if cases with hepatitis B symptoms and/or ALT abnormalities have HBsAg positive test results within 6 months, the case should be diagnosed as acute hepatitis B. If the HBsAg positive result was tested 6 months ago, the case should be diagnosed as chronic hepatitis B. When the tested time of the HBsAg positive result was unknown, and the anti-HBc IgM was positive, the case could be diagnosed as acute hepatitis B. If a case with HBsAg positive test results had no hepatitis B symptoms and no ALT abnormalities, the case should be diagnosed as hepatitis B carrier. We added this information on line 122 in the Methods in page 5.

2.Were the 427 inpatients reported admitted for an HBV related disease? Did outpatients seek medical care for an HBV related disease?

Of the 427 hospitalized cases, we surveyed 400 cases successfully. We did not find the other 27 inpatients’ medical record. Of the 400 inpatients surveyed, only 84 admitted for a HBV related disease, 159 admitted for Gynecologic diseases and in-hospital delivery, 37 were for injury and surgical diseases, 29 were for disease of respiratory system, others were for stomach, cardiovascular, the neurology system, Skin and the urinary system. In China, all inpatients should test HBV when they were admitted. Outpatients also sought medical care for an HBV related disease. If clinician diagnosed them as hepatitis B, they should also be reported to the NNDRS. In this study, we didn’t evaluate these reported HBV cases from outpatient because they didn’t have detailed medical records information as those inpatients.

3.On study limitations you state that “The study was conducted only among hospitalized cases…” but you included 372 outpatients, can you clarify it?

In this study, there were total 799 cases of hepatitis B was reported from the seven hospitals. Among them, 427 (53%) were inpatients and 372 (47%) were outpatients. Among the 427 inpatients, 400 (94%) were evaluated for accuracy of diagnosis because they have detailed medical records information including laboratory test. All the related results in this manuscript were for the 400 inpatients, not including the 372 outpatients.

4.Did you find any similar study in the literature?

We searched total two similar studies in Chinese Journal of Vaccines and Immunization. One is “Actuality analysis on diagnose and report for hepatitis B in Gansu province” published in 2004, another is “Analysis on Reported Hepatitis B Cases on Pilot Surveillance in 18 Counties of 8 Provinces of China” published in 2007. Our study focused on the Gannan prefecture where had the highest reported hepatitis B incidence in Gansu province. We cited these two papers as 8th and 10th reference in this manuscript.

5.I would be nice if you can find any predictor for misclassification such as level of expertise, age, years of practice, subspeciality of clinicians making those reports and for those taking the interviews.

We calculated the inaccurately reporting rates in different departments. The highest inaccurately report rate were 100% in pediatrics, followed by obstetrics and gynecology (91%), traditional Chinese medicine (81%), surgery(74%) and internal medicine(74%). Among 59 duplicated reports, 93%(55/59) were reported from internal medicine department. Among 261 hepatitis B carriers, 47%(146/261) were reported by gynaecology and obstetrics department. We added this information and the following table on line 216 in page8.

Tabble1. The inaccurately reporting rates in different departments

Departments Cases should not be reported Should be reported The rate of inaccurately report（%）

 HepatitisB carriers Duplicate reports Without hepatitis B Total 

Internal medicine 68 55 3 44 170 74

Surgery 32 2 1 12 47 74

Traditional Chinese medicine 13 0 0 3 16 81

Gynaecology and obstetrics 146 1 2 14 163 91

Paediatrics 2 1 1 0 4 100

Total 261 59 7 73 400 82

The mean score was highest in clinicians from department of pediatrics（4.33±2.44）, followed by internal medicine department (4.00±1.39), obstetrics and gynecology department (3.73±2.63), the lowest mean score was 2.66±1.85 for clinicians from clinical laboratory department. The mean score of residents, attending physicians, and associate chief physicians, were 3.79 ± 2.08,3.63 ± 2.58, and 4.5 ± 0.98, respectively. The mean score in clinicians with high school degree were 3.40 ± 2.35, college and bachelor were 3.89 ± 2.11. From the length of service of surveyed clinicians, Clinicians with 20 years had the highest score (4.13 ± 1.73), followed by clinicians with less than 10 years (3.85 ± 1.78) and 10-19 years (3.00 ± 3.20). We added this information on line 248 in page10.

6.I suggest adding in the discussion section how these findings may impact in clinical studies performed for data obtained from databases relying only in medical reports not confirmed by serological data.

In this study, we evaluated the accuracy of clinicians' diagnosis and reporting of hepatitis B by reviewing medical records of inpatients. We made the decision based on the previous disease history, clinical symptoms, imaging examination, laboratory serological test results recorded in medical records. All these data were accurate and true. So our findings could show the real status of clinicians' diagnosis and reporting of hepatitis B. We added this information on line 270 in page 11.

7.Why do you think there is such a high rate of misdiagnosis in an endemic HBV region? Please add some comments in the discussion section.

There were some reasons for such a high rate of misdiagnosis in the endemic HBV region. First, some clinicians did not understand the diagnosis criteria correctly. The hospital does not organize the training on hepatitis B diagnosis; Second, some testing items conducive to the classified diagnosis of hepatitis B are not carried out in local hospitals, such as Anti-HBc IgM, HBV DNA and liver needle biopsy, perhaps because the classified diagnosis has little significance to the treatment of patients. We added this information on line274 in discussion part in page 11.

---

## [Decision Letter · Decision Letter 1]

11 Jul 2022

PONE-D-22-06387R1Accuracy of hepatitis B disease surveillance, Gannan Prefecture, Gansu Province, China; 2017PLOS ONE

Dear Dr. Zhang,

Thank you for submitting your manuscript to PLOS ONE. After careful consideration, we feel that it has merit but does not fully meet PLOS ONE’s publication criteria as it currently stands. Therefore, we invite you to submit a revised version of the manuscript that addresses the points raised during the review process.

Please check and address the last issue raised by Rev#1.

We look forward to receiving your revised manuscript.

Kind regards,

Antonio De Vincentis

Academic Editor

PLOS ONE

Journal Requirements:

Reviewers' comments:

Reviewer's Responses to Questions

**Comments to the Author**

1. If the authors have adequately addressed your comments raised in a previous round of review and you feel that this manuscript is now acceptable for publication, you may indicate that here to bypass the “Comments to the Author” section, enter your conflict of interest statement in the “Confidential to Editor” section, and submit your "Accept" recommendation.

Reviewer #1: All comments have been addressed

Reviewer #2: All comments have been addressed

2. Is the manuscript technically sound, and do the data support the conclusions?

Reviewer #1: (No Response)

Reviewer #2: Yes

3. Has the statistical analysis been performed appropriately and rigorously? 

Reviewer #1: (No Response)

Reviewer #2: Yes

4. Have the authors made all data underlying the findings in their manuscript fully available?

Reviewer #1: (No Response)

Reviewer #2: Yes

5. Is the manuscript presented in an intelligible fashion and written in standard English?

Reviewer #1: (No Response)

Reviewer #2: Yes

6. Review Comments to the Author

Reviewer #1: The authors addressed the comments from reviewers and the manuscript has been modified almost properly. The revised manuscript includes supporting information, but it is completely same as the tables in the main manuscript. Therefore, the supporting information part should be deleted.

Reviewer #2: Thanks to the authors for addressing all comments. Interesting results. I hope you establish an educating campaign in these hospitals about HBV diagnosis.

7. PLOS authors have the option to publish the peer review history of their article (what does this mean?). If published, this will include your full peer review and any attached files.

Reviewer #1: No

Reviewer #2: No

---

## [Author Response · Author response to Decision Letter 1]

1 Sep 2022

Dear editor,

Thanks for your and the reviewers’ valuable comments and recommendations. We had re-checked all the references in the article and replaced reference number 1 with the latest one. We corrected some authors’ name spelling and the journal spelling according to PLOS journal demand. The URL for some references were updated, and some missing literature DOI addresses were added.

We deleted the supporting information part because it was the same as the tables in the main manuscript.

For the second reviewer‘s comments, we recommended the hospitals to conduct education campaign about HBV diagnosis and reporting criteria in the discussion part in the main manuscript. 

Thanks again for all your comments.

Best regards,

Lijie Zhang, China CDC

Correspondent author

---

## [Editor Report · Decision Letter 2]

4 Sep 2022

Accuracy of hepatitis B disease surveillance, Gannan Prefecture, Gansu Province, China; 2017

PONE-D-22-06387R2

Dear Dr. Zhang,

We’re pleased to inform you that your manuscript has been judged scientifically suitable for publication and will be formally accepted for publication once it meets all outstanding technical requirements.

Kind regards,

Antonio De Vincentis

Academic Editor

PLOS ONE
---

## [Editor Report · Acceptance letter]

11 Sep 2022

PONE-D-22-06387R2 

Accuracy of hepatitis B disease surveillance, Gannan prefecture, Gansu province, China; 2017 

Dear Dr. Zhang:

I'm pleased to inform you that your manuscript has been deemed suitable for publication in PLOS ONE. Congratulations! Your manuscript is now with our production department. 

Kind regards, 

on behalf of

Dr. Antonio De Vincentis 

Academic Editor

PLOS ONE